# Antimicrobial resistance prevalence in bloodstream infection in 29 European countries by age and sex: An observational study

Naomi R. Waterlow[1], Ben S. Cooper[2], Julie V. Robotham[3], Gwenan Mary Knight[1,4]*

1 Centre for Mathematical Modelling of Infectious Diseases, Department of Infectious Disease Epidemiology, EPH, London School of Hygiene and Tropical Medicine, London, United Kingdom, 2 Centre for Tropical Medicine and Global Health, Nuffield Department of Medicine, University of Oxford, Oxford, United Kingdom, 3 NIHR Health Protection Research Unit in Healthcare Associated Infections and Antimicrobial Resistance at University of Oxford in Partnership with the UK Health Security Agency, Oxford, United Kingdom, 4 AMR Centre, London School of Hygiene and Tropical Medicine, London, United Kingdom

* gwen.knight@lshtm.ac.uk

## Abstract

### Background

Antibiotic usage, contact with high transmission healthcare settings as well as changes in immune system function all vary by a patient's age and sex. Yet, most analyses of antimicrobial resistance (AMR) ignore demographic indicators and provide only country-level resistance prevalence values. This study aimed to address this knowledge gap by quantifying how resistance prevalence and incidence of bloodstream infection (BSI) varied by age and sex across bacteria and antibiotics in Europe.

### Methods and findings

We used patient-level data collected as part of routine surveillance between 2015 and 2019 on BSIs in 29 European countries from the European Antimicrobial Resistance Surveillance Network (EARS-Net). A total of 6,862,577 susceptibility results from isolates with age, sex, and spatial information from 944,520 individuals were used to characterise resistance prevalence patterns for 38 different bacterial species and antibiotic combinations, and 47% of these susceptibility results were from females, with a similar age distribution in both sexes (mean of 66 years old). A total of 349,448 isolates from 2019 with age and sex metadata were used to calculate incidence. We fit Bayesian multilevel regression models by country, laboratory code, sex, age, and year of sample to quantify resistant prevalence and provide estimates of country-, bacteria-, and drug-family effect variation. We explore our results in greater depths for 2 of the most clinically important bacteria–antibiotic combinations (aminopenicillin resistance in *Escherichia coli* and methicillin resistance in *Staphylococcus aureus*) and present a simplifying indicative index of the difference in predicted resistance between old (aged 100) and young (aged 1). At the European level, we find distinct patterns in resistance prevalence by age. Trends often vary more within an antibiotic family, such as

**Data Availability Statement:** Patient level data is available upon request from the European Antimicrobial Resistance Surveillance Network (EARS-Net) from the Surveillance System (TESSy) for those who meet the criteria for access to confidential data. https://www.ecdc.europa.eu/en/publications-data/european-surveillance-system-

tessy. All code is available in an online repository: https://github.com/gwenknight/ecdc_data_age_sex.

**Funding:** GMK was supported by Medical Research Council UK, https://www.ukri.org/opportunity/career-development-award/ (MR/W026643/1). JVR and BSC were supported by the National Institute for Health Research (NIHR) Health Protection Research Unit in Healthcare Associated Infections and Antimicrobial Resistance (NIHR200915), a partnership between the UK Health Security Agency (UKHSA) and the University of Oxford. The funders had no role in study design, data collection and analysis, decision to publish, or preparation of the manuscript.

**Competing interests:** The authors have declared that no competing interests exist.

**Abbreviations:** AMR, antimicrobial resistance; AST, antimicrobial susceptibility testing; BSI, bloodstream infection; COVID-19, Coronavirus Disease 2019; DRI, drug-resistant infection; EARS-Net, European Antimicrobial Resistance Surveillance Network; ECDC, European Centre for Disease Prevention and Control; MRSA, methicillin-resistant *Staphylococcus aureus*; WHO, World Health Organisation.

fluroquinolones, than within a bacterial species, such as *Pseudomonas aeruginosa*. Clear resistance increases by age for methicillin-resistant *Staphylococcus aureus* (MRSA) contrast with a peak in resistance to several antibiotics at approximately 30 years of age for *P. aeruginosa*. For most bacterial species, there was a u-shaped pattern of infection incidence with age, which was higher in males. An important exception was *E. coli*, for which there was an elevated incidence in females between the ages of 15 and 40. At the country-level, subnational differences account for a large amount of resistance variation (approximately 38%), and there are a range of functional forms for the associations between age and resistance prevalence. For MRSA, age trends were mostly positive, with 72% ($n = 21$) of countries seeing an increased resistance between males aged 1 and 100 years and a greater change in resistance in males. This compares to age trends for aminopenicillin resistance in *E. coli* which were mostly negative (males: 93% ($n = 27$) of countries see decreased resistance between those aged 1 and 100 years) with a smaller change in resistance in females. A change in resistance prevalence between those aged 1 and 100 years ranged up to 0.51 (median, 95% quantile of model simulated prevalence using posterior parameter ranges 0.48, 0.55 in males) for MRSA in one country but varied between 0.16 (95% quantile 0.12, 0.21 in females) to −0.27 (95% quantile −0.4, −0.15 in males) across individual countries for aminopenicillin resistance in *E. coli*. Limitations include potential bias due to the nature of routine surveillance and dependency of results on model structure.

## Conclusions

In this study, we found that the prevalence of resistance in BSIs in Europe varies substantially by bacteria and antibiotic over the age and sex of the patient shedding new light on gaps in our understanding of AMR epidemiology. Future work is needed to determine the drivers of these associations in order to more effectively target transmission and antibiotic stewardship interventions.

### Author summary

#### Why was this study done?

- Antimicrobial resistance (AMR) is a major global public health threat, but little is known about how the prevalence of resistance varies with age and sex.

#### What did the researchers do and find?

- We explored patterns of resistance prevalence and incidence by age and sex in routinely collected data from bloodstream infections (BSIs) across Europe for 8 bacterial species.

- We fitted a Bayesian multilevel regression model to quantify the variation nationally and subnationally.

- Distinct patterns in resistance prevalence by age were observed across Europe for different bacteria.

- Sex was often only weakly associated with resistance, except across ages in *Escherichia coli* and *Klebsiella pneumoniae*, and at younger ages for *Acinetobacter* species, where it was higher in males.

### What do these findings mean?

- These findings highlight important gaps in our knowledge of the epidemiology of AMR that are difficult to explain through known patterns of antibiotic exposure and health-care contact.

- Differences in AMR burden by age and sex may be explained by cultural differences between countries as well as variation in pathways to infection between bacteria.

- Our findings suggest that there may be value in considering interventions to reduce AMR burden that take into account important variations in AMR prevalence with age and sex.

- This study is limited by the nature of routine surveillance, the lack of open availability of disaggregated data, and the model structures explored.

## Introduction

Antimicrobial resistance (AMR) is a global public health priority [1]. Understanding how it will be affected by the dramatic demographic shifts that are underway worldwide is a key knowledge gap. The World Health Organisation (WHO) has estimated that 1 in 5 people in the world will be aged 60 years or older by 2050 [2]. Incidence of bacterial infections is known to increase by age [3] and vary by sex [4]. The higher burden of infection in older age groups [5,6], results in higher antibiotic exposure and higher contact with healthcare settings which are known hotspots of resistant bacteria transmission. However, there is not a simplistic increase in resistance in all pathogens by age. Determining how the above interact to drive the dynamics of drug-resistant infections (DRIs) is a vital part of understanding how best to tackle AMR.

Age- and sex-disaggregated data are collected by most routine AMR surveillance schemes. The WHO Global Antimicrobial Resistance and Use Surveillance System (GLASS) requests age- and sex-stratifications from reporting countries [7]. However, this data are not currently openly available at low age-band segregation (i.e., more than 4 broad categories) with sex—for example, not from the WHO GLASS dashboard [8] nor the European Centre for Disease Prevention and Control (ECDC) ATLAS dashboard [9] nor the US Centres for Disease Control and Prevention [10]. The recent WHO reports also have not presented analysis of how resistance prevalence varies by these demographic factors [11]. The dramatic, often exponential, increase in infection incidence with older age has been reported in several places [12–15] as well as the differences by sex [16]. However, how this burden is split into resistant or susceptible infection by patient age and sex is relatively rarely reported. Multiple attempts to predict AMR burden are hampered by a basic lack of surveillance data and yet factors such as sex and age are variables that are nearly always available for analysis.

The importance of age being linked to variation in AMR has been graphically explored before for Europe [17] and more comprehensively in single setting studies (e.g., [18–20]). Complex statistical analysis based on the Global Burden of Disease methods have produced age- and sex-specific estimates of mortality rates by European country attributable to all AMR [21]. However, to our knowledge there has been no comprehensive analysis of the relationship between age and AMR in infection between bacteria across multiple countries. This is despite the wide awareness of age-specific effects for infection that have only been emphasised by the Coronavirus Disease 2019 (COVID-19) pandemic [22].

Despite sex being a well-established risk factor for specific bacterial infections such as urinary tract infections, how prevalence of DRI varies between the sexes (and genders) is vastly underexplored in the literature [23]. This is despite many studies of infections caused by specific bacterial pathogens or syndromes finding a difference in resistance prevalence in infection between the sexes [24–29]. In 2018, the WHO called for countries to take the first step to better considering "gender and equity" in National Action Plans for AMR [30], which have historically lacked such considerations (e.g., in Southeast Asia [31]).

Prevalence of resistance in infection is known to vary between countries [8,11] and subnationally, by factors such as deprivation level [32–34]. This may be linked in part to differences in healthcare structures and antibiotic usage [35,36]. Other national level healthcare structures and cultural differences are likely to have wider impacts on AMR patterns by age and sex. For example, variation in birth rates by age between countries [37], as well as type of birth (vaginal versus cesarean) [38] will impact the type of antibiotic as well as healthcare exposures in women. Determining how these cultural factors interplay with biological factors as we age and across sexes is key to understanding the nuanced interventions required to tackle AMR.

Here, we use a large dataset of routinely collected information on bloodstream infections (BSIs) to explore trends in prevalence of antibiotic resistance and infection by age and sex across Europe.

## Methods

### Ethics statement

This work to analyse routinely collected data was approved by the London School of Hygiene and Tropical Medicine ethics board (ref 28157).

### Data

We analysed the European Antimicrobial Resistance Surveillance Network (EARS-Net) patient level data for 2015 to 2019 reported to European Centre for Disease Prevention and Control (ECDC) by Austria, Belgium, Bulgaria, Cyprus, Czechia, Germany, Denmark, Estonia, Greece, Spain, Finland, France, Croatia, Hungary, Ireland, Iceland, Italy, Luxembourg, Latvia, Malta, the Netherlands, Norway, Poland, Portugal, Romania, Sweden, Slovenia, Slovakia, and the United Kingdom [39,40]. Countries were anonymised using a random 3-letter code, which is used throughout the paper as data at this level of detail by age and sex is not publicly available, because the required national level permissions are not in place [39]. EARS-Net collects routine clinical antimicrobial susceptibility testing (AST) results, alongside some patient data, including sex and age, from EU/EEA countries (we use the term European throughout). The general quality and comparability of the data is evaluated through a standard annual external exercise [41] with the AST results taken from shared protocols [40,42]. The data consists of AST for the first blood and/or cerebrospinal fluid isolate (<0.7% of this dataset) of every patient with an invasive infection associated with one of the pathogens under surveillance (Section 2 in S1 Appendix). Levels of coverage are discussed and explored in the calculation of

incidence (see below and Section 3 in S1 Appendix). In our main analysis, we exclude individuals aged 0, due to their stark difference in immune dynamics, linked in part to waning maternal antibodies [43,44], and healthcare contact patterns of this subset of critically ill children, but run a sensitivity analysis including them.

Individual patient data from EARS-Net was extracted with information on the age and sex of the patient, resistance presence, laboratory code, year of sample, and reporting country. For resistance prevalence calculations, we used the susceptibility test result data for 2015 to 2019 in those aged one or older, with data on age and sex. We analysed missing data both in terms of (a) distribution of age and sex within those not tested for resistance; and (b) resistance prevalence in those without age and sex information. For incidence calculations, we included all isolates with recorded age and sex values, for those aged one or older.

We used the United Nations subregion definitions, except for Cyprus, which was grouped with Southern Europe (instead of being the only Western Asia country). Some susceptibility data grouped results for multiple antibiotics together: "aminopenicillins" are ampicillin or amoxicillin, "3G cephalosporins" are cefotaxime, ceftriaxone or ceftazidime, "fluoroquinolones" are ciprofloxacin, levofloxacin or ofloxacin, "aminoglycosides" are gentamicin or tobramycin, "macrolides" are azithromycin, clarithromycin or erythromycin, "penicillins" are penicillin or oxacillin, "carbapenems" are imipenem/meropenem (Table 1 in ECDC reports [45]). Where we had multiple susceptibility results for individual antibiotic within a drug family (beta-lactam), we grouped antibiotics by AWaRe classifications [46]. We follow the ECDC analysis and assume "sex" rather than "gender" was recorded in the data.

1.  *Prevalence of resistance in infection by age and sex*
    Using the cleaned data, we explored variation in patterns in aggregated sex- and age-based resistance prevalence in infection at the European and subregional levels.

2.  *Incidence of infection by age*
    Following the methods of Cassini and colleagues [47] (Sections 3 and 4 in S1 Appendix), it was assumed that all eligible invasive isolates are reported by the participating laboratories. The estimated coverage of these laboratories was then used as an inflation factor to calculate the number of BSIs. Data for country coverage was taken from previous EARS-Net reports and the Cassini and colleagues estimates for 2015, 2018, 2019, and 2020. The incidence of infection in each of these years was calculated by dividing the number of isolates from patients in each 5-year age and sex band by the corresponding population sizes from the World Bank DataBank [48], up to a pooling of all those aged 80 or older. We report an estimated incidence for 2019.

3.  *Trend analysis for resistance proportion by age*
    Multilevel regression models were fitted to the ECDC data to understand the impact of including age and sex in models of resistance prevalence. We used a Bayesian framework using the *R* package *brms* [49] and ran models using the No U-turn Sampling separately for each bacteria-antibiotic combination, using data from 2015 to 2019. Individual-level data was aggregated to group level by country, laboratory code, sex, age, and year of sample and standardised as appropriate (Section 5 in S1 Appendix). Models were considered converged if the Rhat was <1.1, a sufficient Effective Sample Size for each parameter was reached and we checked for divergent transitions (Section 5 in S1 Appendix). We initially ran 3,000 iterations and extended this to 5,000 for those models that had not reached convergence at this point. Country and laboratory code were included as substantial variation was observed between them under a variance-components model (Section 5 in S1 Appendix). Only the

**Table 1. Baseline characteristics of data included for exploration of resistance prevalence in BSIs across Europe for 15 antibiotics across 8 bacteria in 29 countries for 2015–2019.** The split of the susceptibility results by country is given in Table A2 in S1 Appendix. The definition of an "antibiotic" was linked to the data given so is at different levels (e.g., separate aminoglycosides were included as well as an antibiotic category of "aminoglycoside"). Fluroquinolones resistance was labelled the same across species though there were species-specific definitions. Age given in years. MRSA covers oxacillin and cefotoxin. 3G = third-generation. Pip-taz = piperacillin-tazobactam. SD = standard deviation.

| Characteristics | | All | Female | Male |
|---|---|---|---|---|
| **Susceptibility results (*n*(%))** | | 6,862,577 | 3,211,521 (47%) | 3,651,056 (53%) |
| **Number of patients** | | 944,520 | 444,778 (47%) | 538,723 (53%) |
| **Age (mean, SD)** | | 66 (19.6) | 66 (20.6) | 66 (18.7) |
| **Number of susceptibility results Range (mean; SD)** | **Country (*n* = 29)** | 8,391–1,137,670 (236,641; 272,901) | 4,268–547,018 (110,742; 128,611) | 3,989–590,652 (125,898; 144,594) |
| | **Bacteria (*n* = 8)** | 82,085–4,032,238 (857,822; 1,313,543) | 35,084–2,122,471 (401,440; 703,758) | 47,001–1,909,767 (456,382; 613,016) |
| | **Antibiotic (*n* = 15)** | 74,342–1,184,265 (457,505; 359,541) | 26,635–545,803 (214,101; 176,634) | 41,597–638,462 (243,404; 183,812) |

| | | | **Total number of susceptibility results** | | |
|---|---|---|---|---|---|
| **Bacteria** | **Antibiotic** | **Age (mean)** | **All** | **Female** | **Male** |
| *Acinetobacter* species | Amikacin | 61 | 15,298 | 6,491 | 8,807 |
| | Aminoglycosides | 61 | 22,174 | 9,506 | 12,668 |
| | Carbapenems | 61 | 22,329 | 9,543 | 12,786 |
| | Fluroquinolones | 61 | 22,284 | 9,544 | 12,740 |
| *Enterococcus faecalis* | Aminopenicillins | 69 | 89,517 | 29,780 | 59,737 |
| | High-level aminoglycoside | 68 | 57,831 | 19,731 | 38,100 |
| | Vancomycin | 69 | 91,995 | 30,585 | 61,410 |
| *Enterococcus faecium* | Aminopenicillins | 67 | 59,674 | 23,444 | 36,230 |
| | High-level aminoglycoside | 67 | 36,471 | 14,314 | 22,157 |
| | Vancomycin | 67 | 61,855 | 24,252 | 37,603 |
| *Escherichia coli* | Amikacin | 67 | 349,169 | 182,658 | 166,511 |
| | Aminoglycosides | 67 | 618,839 | 326,059 | 292,780 |
| | Aminopenicillins | 67 | 532,227 | 280,669 | 251,558 |
| | Carbapenems | 67 | 604,618 | 318,300 | 286,318 |
| | 3G cephalosporins | 67 | 612,331 | 322,977 | 289,354 |
| | Ertapenem | 67 | 264,862 | 140,223 | 124,639 |
| | Fluoroquinolones | 67 | 619,648 | 326,594 | 293,054 |
| | Pip-taz. | 67 | 430,544 | 224,991 | 205,553 |
| *Klebsiella pneumoniae* | Amikacin | 66 | 96,924 | 37,872 | 59,052 |
| | Aminoglycosides | 67 | 148,410 | 58,048 | 90,362 |
| | Carbapenems | 67 | 146,551 | 57,284 | 89,267 |
| | 3G cephalosporins | 67 | 148,192 | 57,977 | 90,215 |
| | Ertapenem | 67 | 67,062 | 26,248 | 40,814 |
| | Fluoroquinolone | 67 | 149,122 | 58,325 | 90,797 |
| | Pip-taz. | 68 | 108,092 | 41,909 | 66,183 |
| *Pseudomonas aeruginosa* | Amikacin | 66 | 59,968 | 21,797 | 38,171 |
| | Aminoglycoside | 67 | 76,015 | 27,245 | 48,770 |
| | Carbapenem | 67 | 76,055 | 27,251 | 48,804 |
| | Ceftazidime | 67 | 74,342 | 26,635 | 47,707 |
| | Fluoroquinolone | 67 | 75,944 | 27,236 | 48,708 |
| | Pip-taz. | 67 | 73,729 | 26,430 | 47,299 |
| *Staphylococcus aureus* | Fluoroquinolone | 64 | 258,605 | 97,323 | 161,282 |
| | MRSA | 64 | 286,731 | 107,916 | 178,815 |
| | Rifampicin | 64 | 232,585 | 87,379 | 145,206 |

*(Continued)*

**Table 1.** (Continued)

| Characteristics | | All | | Female | Male |
|---|---|---|---|---|---|
| *Streptococcus pneumoniae* | 3G cephalosporins | 63 | 56,908 | 25,704 | 31,204 |
| | Fluoroquinolone | 63 | 58,662 | 26,781 | 31,881 |
| | Macrolide | 63 | 79,731 | 36,814 | 42,917 |
| | Penicillins | 63 | 77,283 | 35,686 | 41,597 |

BSI, bloodstream infection; MRSA, methicillin-resistant *Staphylococcus aureus*.

sexes "male" and "female" were included in the analysis and records missing age or sex were dropped.

Thus, for each bacteria-antibiotic combination, our data consisted of multiple groupings of individual samples of a bacterium tested for resistance to that antibiotic. Each grouping $i$ had a unique combination of country (c), laboratory code (l), sex, age, and year of sample and hence a linked number of samples ($n$) and proportion resistant ($p$).

For each bacteria-antibiotic combination, we ran a multilevel logistic regression model to predict the probability of an isolate being resistant to the antibiotic, assuming a binomial distribution over the number of samples in each grouping. Our model included both age and sex terms (Eqs 1 and 2).

$$y_i \sim Binomial(n_i, p_i) \tag{1}$$

$$p_i = \beta_0 + \beta_t * t_i + \beta_a * age_i + \beta_{a^2} * age^2_i + \beta_g * sex_i + \beta_{ag} * age_i * sex_i + v_{c(i)} + v_{c(i)a(i)} * age_i + u_{c(i),l(i)} + \epsilon_i \tag{2}$$

Where $y$ is the resistance variable, taking a value of 0 or 1, $n$ is the number of samples, and $p$ the probability of the sample being found to be resistant (NAs were excluded, Section 5 in S1 Appendix). The subscripts $c$, $l$, and $i$ denote country, laboratory code, and grouping level. $\beta_0$ is the overall intercept, $\beta_t$ is slope coefficient for time, and $t_i$ is year. $\epsilon_i$ is the residual error, $u_{c(i),l(i)}$ is the level-2 random error on laboratory code, and $v_{c(i)}$ is the level-3 random error on country. $\beta_a$ is the age effect coefficient, $\beta_{a^2}$ is the age squared effect coefficient, and $v_{c(i)a(i)}$ is the country-level age effect coefficient. $\beta_g$ is the sex effect coefficient and $\beta_{ag}$ is the sex and age interaction coefficient. The sex variable takes a value of 0 or 1, being 1 for males. We chose to include an $age^2$ explanatory variable, as previous analysis had identified nonlinear trends with age and antibiotic use (a key driver of resistance) is known to have nonlinear, often quadratic relationship with age [50].

All random errors are assumed to be normally distributed, and we assume the default priors on all covariates in the main analysis from the *brms* package [49], but run a sensitivity analysis with weakly informative regularising priors.

To determine an overall impact of age for each bacteria-antibiotic combination and country, we calculated the difference in the model-predicted proportion resistant between young individuals (aged 1) and older individuals (those aged 100), using the posterior predictions from the model fit. We did this across all posterior samples, from which we calculated the median and 95% quantiles. This simplifying index was chosen to capture and illustrate one aspect of the differences seen (the change between young and old). We explore the robustness of this index to different definitions of young (age 1 to 20) and older (age 50 to 100).

## Sensitivity analysis

We explored further data disaggregation of incidence by patient location when the sample was taken (inpatient versus outpatient and the hospital unit or ward type, e.g., haematology or emergency department). For incidence analysis, we explored varying the inflation factor for the incidence of infection to check robustness of age and sex patterns.

For the modelling analysis, we explored including samples from individuals aged 0, including regularising priors and using a model selection-based approach. These sensitivity analyses were run for MRSA.

## Results

Our analysis was in 3 stages. Firstly, we explored the trends in resistance prevalence by age and sex across Europe. Secondly, we estimated and quantified the incidence of infection for each of the bacterial species by age and sex. Thirdly, we quantified the proportion of those infections that were due to resistant bacteria for different bacteria-antibiotic combinations by age and sex, country and subnational indicator (laboratory) by fitting multilevel models. We exemplify the outputs of the multilevel modelling by exploring results for 2 examples: aminopenicillin resistance in *E. coli* and methicillin resistance in *S. aureus* chosen for their large contributions (>40%, [51]) to the aetiology of BSIs, high number of samples (Table 1) and associated important resistance in Europe and globally [45,52].

## Data

For the resistance prevalence calculations, we used a total of 6,862,577 susceptibility results (74% of the original available) across 29 European countries for 15 antibiotic groupings in 8 bacteria for 2015 to 2019 (Table 1, Section 2 in S1 Appendix). The average age of the patients with BSIs from whom the samples came was 66 (standard deviation: 19.6), and the majority were male (53%). The age- and sex-distribution was similar across all bacteria and antibiotic groupings. All countries and bacteria were included the analysis despite large variations in the number of susceptibility results reflecting the aetiology of BSIs and population sizes (Table 1, Section 2 in S1 Appendix).

For incidence calculations, we used all isolates with age and sex information taken from patients aged 1 or older (a total of 349,448 isolates in 2019) (Section 3 in S1 Appendix). This was 91% of all isolates, with a range between 5,637 and 154,071 isolates used across Europe in 2019.

## Resistance prevalence: European level

At the European level, there were clear nonlinear differences in the prevalence of resistance in infection by age and sex for different bacteria-antibiotic combinations (Fig 1). These patterns were robust across subregions of Europe (Section 1 in S2 Appendix). However, prevalence of resistance was generally higher in Southern and Eastern Europe, with stronger age-related trends (e.g., for methicillin resistance in *S. aureus* and across *Acinetobacter* species). The age-associated patterns varied more within drug-families than within certain bacteria (patterns within each colour are more different than within each row of Fig 1). For example, patterns of resistance across drug families were highly similar across all antibiotics included for some bacteria such as *Acinetobacter* species (Fig 1A) while there was substantial variation within resistance proportions by age for fluroquinolones (blue data, Fig 1). For some bacterial species, such as *S. pneumoniae* (Fig 1D), resistance trends were similar for subsets of antibiotics. Sex

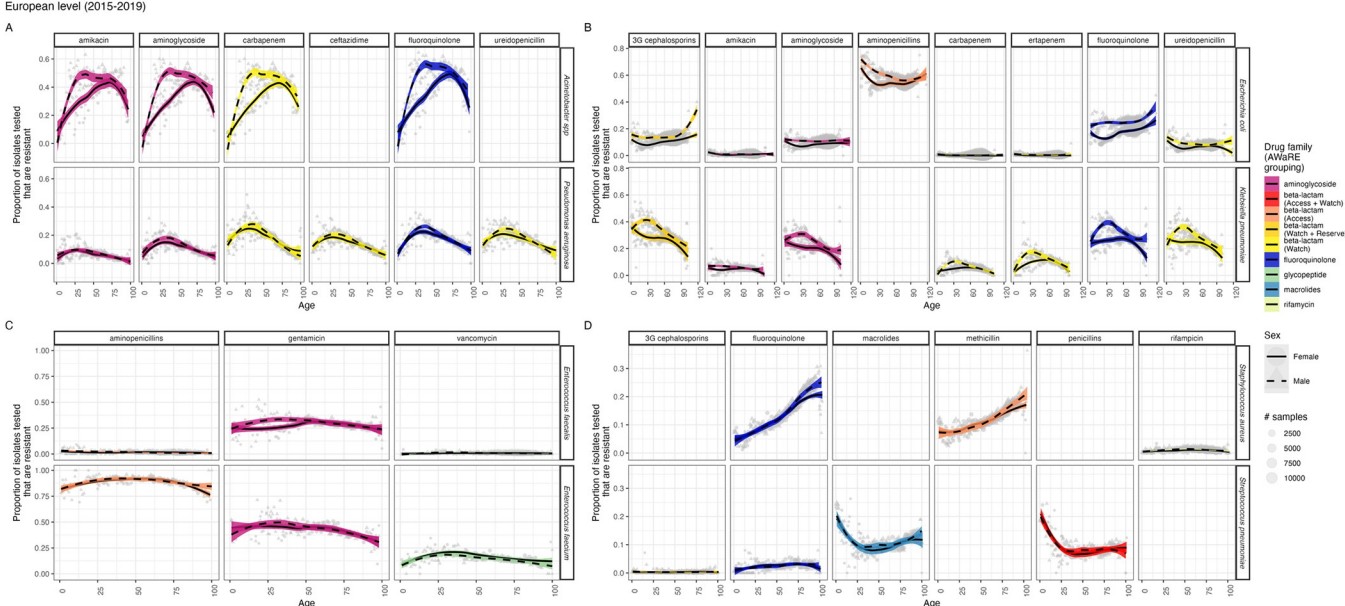

**Fig 1. Trends in resistance prevalence in BSIs vary by antibiotic, bacteria, and demographic factors across Europe.** The proportion of isolates from BSIs tested (y axis) that are resistant to each antibiotic (panel) within drug families and AWaRe groupings (for beta-lactams) (colour) for each bacteria (row) is shown for all European data over 2015–2019 by age (x axis) for the 4 gram-negative (A+B) and 4 gram-positive bacteria (C+D). Data is shown as points with number of samples indicated by size of point. Shaded areas are 95% confidence intervals around the LOESS fit line by sex (linetype). AWaRe groupings were used here to better distinguish clinically important subsets within the beta-lactam family. Blank panels indicate where no data was available. BSI, bloodstream infection; LOESS, locally estimated scatterplot smoothing.

has little impact on many of the age-related trends except for *E. coli* and *K. pneumoniae*, and at younger ages for *Acinetobacter* species (Fig 1B).

## Incidence

As expected, across Europe, BSI incidence was u-shaped for most bacteria, substantially increasing with older ages with clear differences between the sexes (Fig 2). Men had a higher incidence of infections from approximately age 35 onwards, except for *E. coli* between ages 15 and 40 where women had a higher incidence (Fig 2A) and *S. pneumoniae*. These patterns were robust at the country level and over time (Section 2 in S2 Appendix). Differences in infection incidence between the pathogens reflect the overall burden in infection, with ranking incidence rates being (from highest): *E. coli*, *S. aureus*, *K. pneumoniae*, *E. faecium*, *P. aeruginosa*, *E. faecalis*, *S. pneumoniae*, *Acinetobacter* species.

The combination of these age-related trends in number of infections (Fig 2) with those in proportion resistant (Fig 1) lead to exponential increases in the number of resistant infections with age among adults (Section 3 in S2 Appendix) for all bacteria-antibiotic combinations.

## Resistance prevalence: Model results

Our logistic model converged for 34 bacteria-antibiotic combinations (89%) (Section 4 in S1 Appendix) and had substantial effects for at least one of age, age², or the interaction between age and sex (Table 2). Sex had less of a clear importance for many bacteria-antibiotic combinations, with at least one of the sex intercept or interaction terms being substantial for 19 of the 34 bacteria-antibiotic combinations. Full results for all bacteria-antibiotic combinations can be found in S3 Appendix.

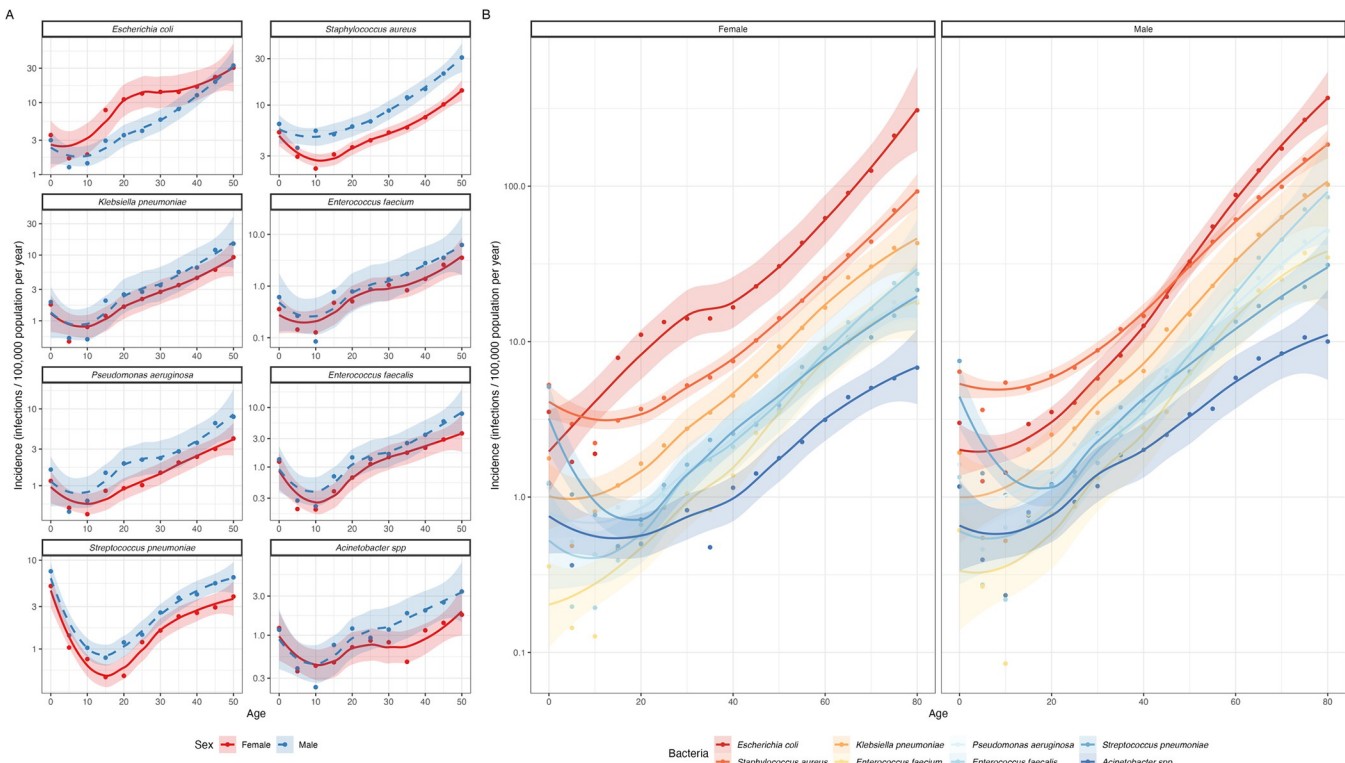

**Fig 2. Incidence of BSIs per 100,000 population in 2019 across European countries for 8 bacterial pathogens.** To demonstrate trends clearly, the incidence is split to show (A) patterns in the first 50 years of life by sex and bacteria and then (B) shows the lifelong trends split by sex (panel) and bacteria (colour). Shaded areas are 95% confidence intervals using an LOESS fit. Infections in individuals younger than 0 are excluded, and those aged 80 and older are pooled into the 80-yr data point. The y axis is on a log scale (base 10). BSI, bloodstream infection; LOESS, locally estimated scatterplot smoothing.

The nonlinear trends in resistance prevalence by age were approximated in the model by a combination of linear and quadratic age effects, with variable combinations of associations leading to varying negative and positive coefficients (Table 2). The decomposition of these effects can be seen in Section 4 in S2 Appendix.

## National and subnational variation in resistance prevalence

We found that resistance prevalence varied substantially by country (as expected) but also within country, with subnational differences accounting for a large amount of resistance variation (approximately 38%, Section 5 in S2 Appendix). There were a range of shapes (both convex and concave) for the associations by sex between age and resistance prevalence (S3 Appendix).

To demonstrate this substantial national and subnational variation, we focus on 2 example bacteria-antibiotic combinations, demonstrating model predictions of subnational (laboratory level) variation for countries across the range of coefficients for age effect (Fig 3) and the country-level differences in resistance between the young and old (Fig 4).

For MRSA, most countries (e.g., for males, 72%, 21/29 countries) have a positive trend with age (driven in part by a high age$^2$ coefficient, Fig 3A), while for aminopenicillin resistance in *E. coli* the age trend is mostly negative (males, 93%, 27/29 countries), with a lower proportion resistant with age (Figs 3B and 4). Country-level effects (panels, Fig 3C and 3D) as well as laboratory (subnational) effects (lines, Fig 3C and 3D) were highly important in capturing proportion resistant by age.

**Table 2. Heatmap of the values of the fixed effect parameters for each bacteria-antibiotic model.** Orange indicates a positive coefficient and blue indicates a negative coefficient (in both cases, where the 95% credible intervals of the posterior parameter estimate do not cross 0). White indicates the coefficient was neither positive nor negative (i.e., posterior credible intervals cross 0). An equivalent table with the parameter values can be found in the supplement (Section 4 in S2 Appendix); (m) indicates that the parameter is the coefficient for males. Fluoroquinolone resistance definitions varied between species (S1 Appendix). MRSA primarily indicates oxacillin or cefoxitin resistance, but other markers are accepted for oxacillin, if oxacillin was not reported. See protocol for details [40].

| Bacteria | Antibiotic | Year | Age | Age² | Sex(m) | Age:sex(m) |
|---|---|---|---|---|---|---|
| *Acinetobacter* species | Amikacin | | orange | blue | | |
| | Aminoglycosides | blue | orange | blue | orange | |
| | Carbapenems | | orange | blue | | |
| | Fluroquinolones | blue | orange | blue | orange | |
| *Enterococcus faecalis* | High-level aminoglycoside | | blue | blue | | |
| | Vancomycin | | blue | blue | | |
| *Enterococcus faecium* | Aminopenicillins | orange | blue | blue | | |
| | High-level aminoglycoside | orange | blue | blue | | |
| *Escherichia coli* | Amikacin | | | orange | orange | blue |
| | Aminoglycosides | blue | | | orange | blue |
| | Aminopenicillins | blue | blue | blue | orange | blue |
| | Carbapenems | orange | blue | | orange | |
| | Fluoroquinolones | blue | orange | blue | orange | blue |
| | Third-generation cephalosporins | orange | orange | | orange | blue |
| | piperacillin-tazobactam | blue | | | orange | blue |
| *Klebsiella pneumoniae* | Amikacin | blue | orange | blue | | |
| | Aminoglycosides | blue | | blue | | orange |
| | Carbapenems | orange | orange | blue | orange | |
| | Ertapenem | orange | orange | blue | orange | |
| | Fluoroquinolones | orange | orange | blue | orange | orange |
| | Third-generation cephalosporins | | | blue | orange | orange |
| | Piperacillin-tazobactam | orange | orange | blue | | orange |
| *Pseudomonas aeruginosa* | Amikacin | blue | orange | blue | | |
| | Aminoglycosides | blue | orange | blue | | |
| | Carbapenems | blue | orange | blue | | |
| | Ceftazidime | | orange | blue | | |
| | Fluoroquinolone | | orange | blue | | orange |
| | Piperacillin-tazobactam | blue | orange | blue | | |
| *Staphylococcus aureus* | Fluoroquinolone | blue | orange | orange | orange | orange |
| | MRSA | blue | blue | orange | orange | orange |
| | Rifampicin | blue | orange | blue | orange | blue |
| *Streptococcus pneumoniae* | Macrolide | blue | blue | orange | | |
| | Penicillins | orange | blue | orange | | |
| | Fluoroquinolone | orange | | | | |

MRSA, methicillin-resistant *Staphylococcus aureus*.

There were important sex effects in both the intercept and age-slope terms for both MRSA and aminopenicillin-resistance in *E. coli* (Fig 3A and 3B), which resulted in clear differences in age-association by sex in these examples (Fig 4).

For MRSA, a higher proportion of samples were predicted to be resistant at older (age 100) than younger (age 1) ages (Fig 4A). The magnitude of this difference varied but reached a maximum difference in proportion of 0.51 (median, 95% quantile 0.48, 0.55, for country PUB) between males aged 1 and 100. For aminopenicillin resistance in *E. coli* for many countries, a

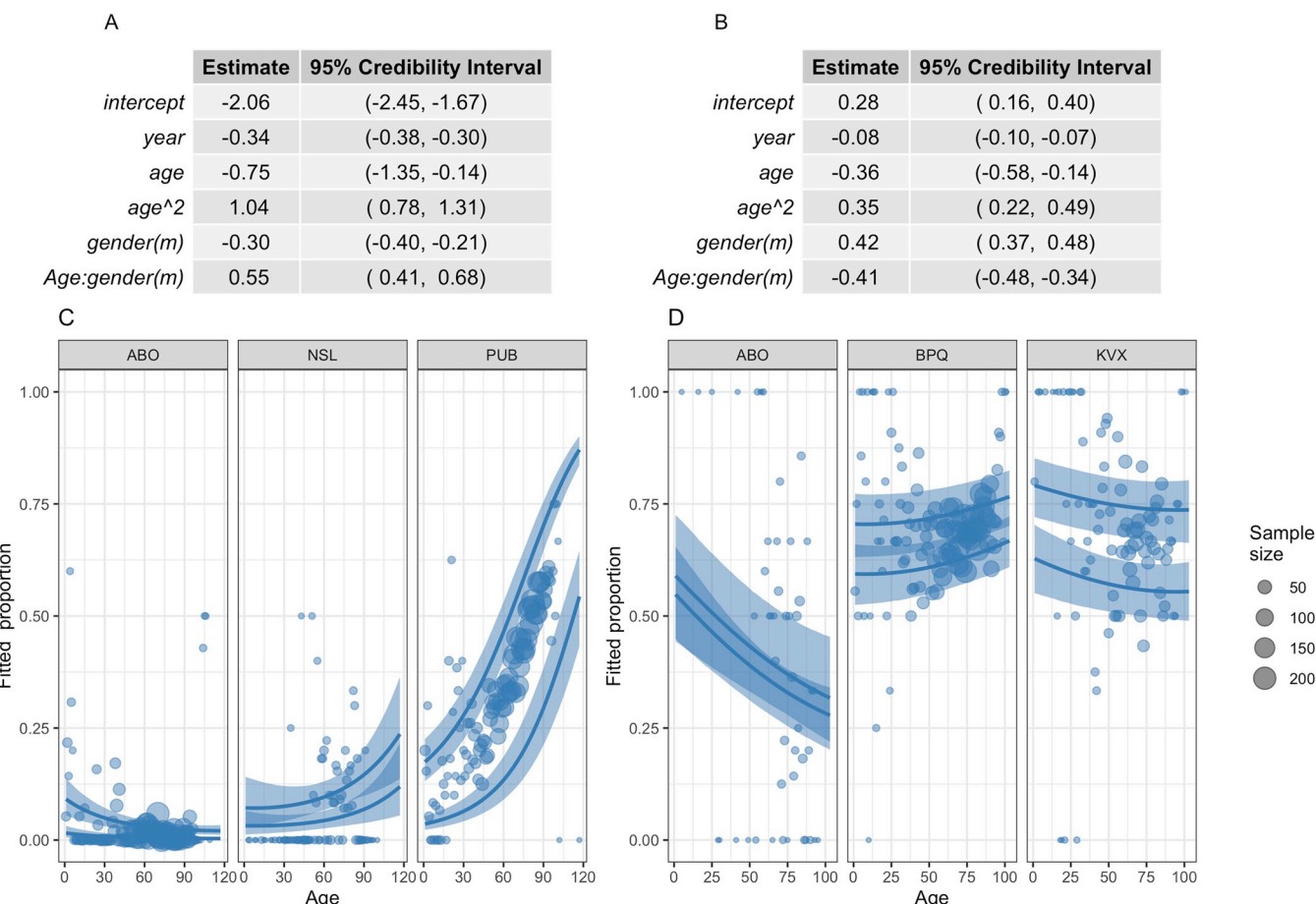

**Fig 3. Model parameters and example model predictions for MRSA (left) and aminopenicillin resistance in *E. coli* (right).** (A, B) Model parameters. (m) indicates that the parameter is the coefficient for males. (C, D) Data (points) and model predictions (lines) with 95% credible intervals (ribbons) for males for the most extreme (left and right panels) and the middle country (middle panel) estimated age slope. Each country has 2 lines, depicting the predictions for the most extreme laboratories in the country. Data sample size (shown by dot size) is grouped across years and laboratories. MRSA primarily indicates oxacillin or cefoxitin resistance, but other markers are accepted for oxacillin, if oxacillin was not reported. See protocol for details [41]. Country labels are a random anonymised 3-letter code used for this study only but consistent across all analyses. MRSA, methicillin-resistant *Staphylococcus aureus*.

lower proportion of samples were predicted to be resistant at age 100 than age 1 (Fig 4B), with the magnitude of the age effect varying from 0.16 (median, 95% quantiles 0.12, 0.21, country BPQ female) to −0.27 (95% quantiles −0.4, −0.15, country ABO, male). The trends in this simplifying index by country were robust to comparing resistance prevalence at ages 50 to 100 to age 1, and at ages 1 to 20 to that at age 100 (Section 6 in S2 Appendix).

## Variations in resistance prevalence with age by country, antibiotic, and bacteria

While the 2 bacteria-antibiotic samples chosen above show substantial trends, there are many bacteria-antibiotic combinations where no age or sex trend is seen, or where there is little similarity within a bacteria-antibiotic combination across countries (completely overlapping confidence intervals when looking at the difference between resistance prevalence at ages 1 to 100) (Section 5 in S2 Appendix). Additional bacteria-antibiotic combinations where a >5% change in resistance proportion between old and young (ages 100 versus 1) for multiple countries was seen include third-generation cephalosporin resistance in *E. coli* and *K. pneumoniae*,

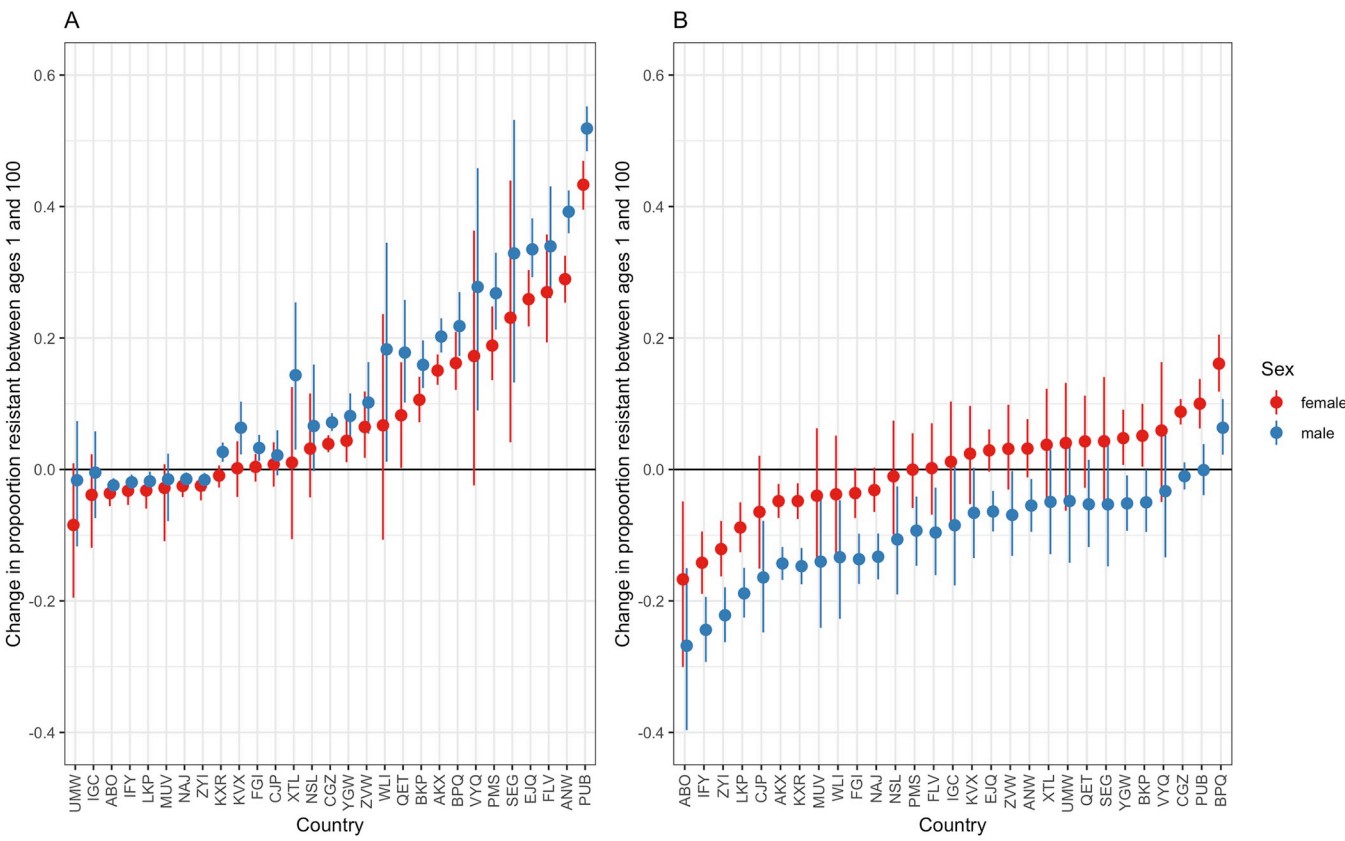

**Fig 4. Change in model-predicted proportion resistant between older (age 100) and younger (age 1) patients.** This index is shown for each country and sex for MRSA (A) and aminopenicillin resistance in *E. coli* (B), with the point indicating the median and the error bars the 95% quantiles across model predictions. Country labels are random anonymised 3-letter code used for this study only but consistent across all analyses. MRSA, methicillin-resistant *Staphylococcus aureus*.

fluroquinolone resistance in *E coli*, *K. pneumoniae*, and *S. aureus*, aminoglycoside resistance in *E. coli* and *K. pneumoniae*, and carbapenem resistance in *P. aeruginosa* (Section 5 in S2 Appendix). The difference in resistance prevalence for the latter is relatively stable at approximately −10% across countries, while others have large variability in magnitude across countries. There is also substantial variation in resistance prevalence between old and young (ages 100 versus 1) between different bacteria-antibiotic combinations within countries.

Within bacterial species by comparing the model-predicted proportion resistant between old and young (ages 100 versus 1), age-related trends were seen across all antibiotics for *Acinetobacter* species (positive, both sexes), *E. coli* (positive, female), and *P. aeruginosa* (negative, both sexes) (Fig 5), although for these the majority were not significant at the 95% level, and have relatively small impacts. No clear age-related trends were seen in our modelling results within Gram stain groupings (Fig 5), nor within antibiotic families (colours in Fig 5), further emphasising the trends seen at the European level (Fig 1).

## Sensitivity analysis

Analysis of infection incidence by patient type and healthcare location when the sample was taken revealed large differences between countries across Europe likely linked to differences in healthcare systems and reporting protocols (Sections 7 and 8 in S2 Appendix). Hence, we could not here explore resistance prevalence further by these differences. Age and sex patterns

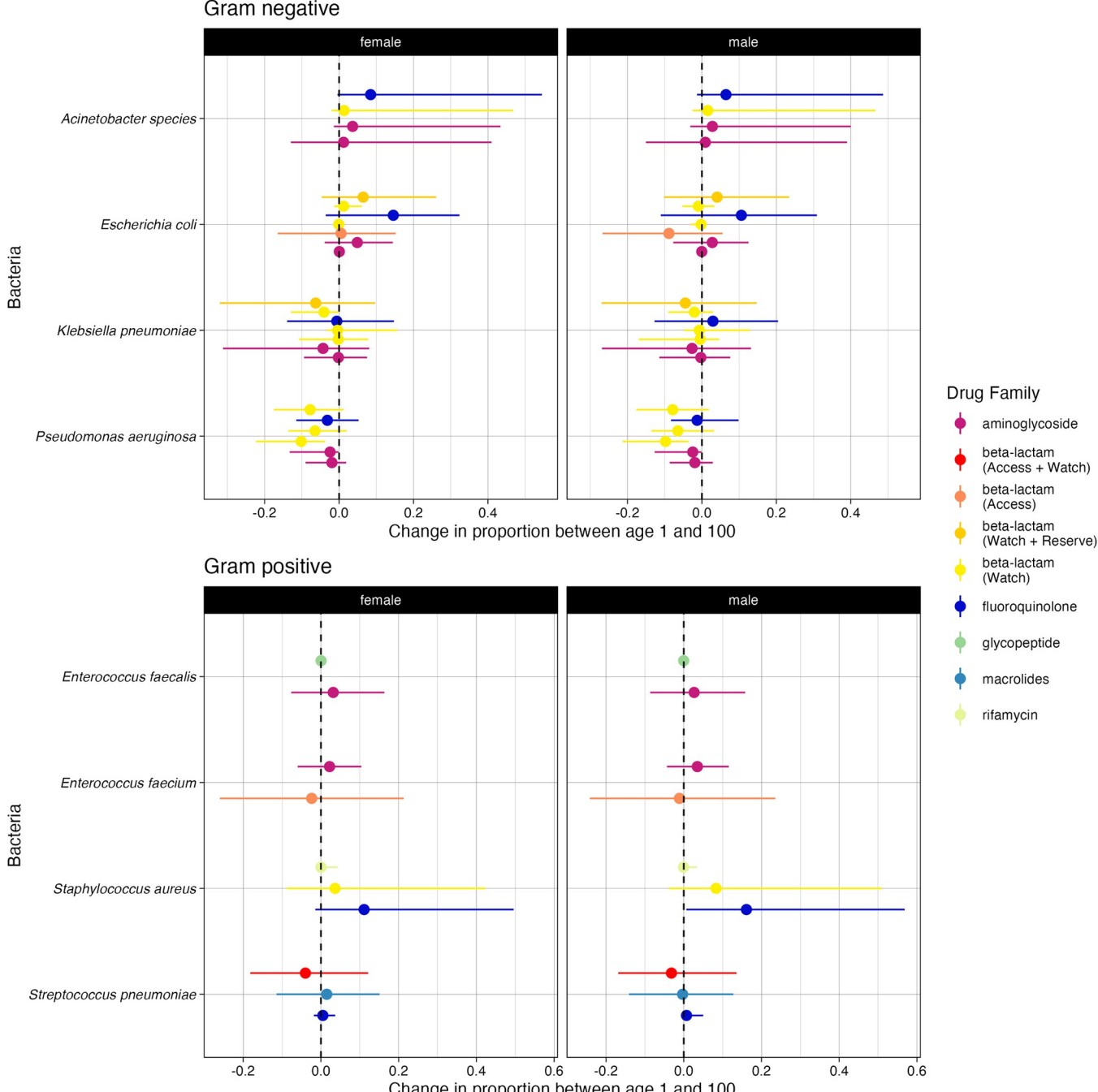

**Fig 5. Change in model-predicted proportion of samples resistant between older (age 100) and younger (age 1) patients with BSIs.** The index is split by sex (panels) and Gram stain test result. Each point indicates the median index for each individual antibiotic coloured by drug family with the error bars showing the 95% quantiles across model predictions. The dashed line indicates 0 (i.e., no difference in model-predicted difference between resistance prevalence at age 1 vs. 100). The anonymised country-level results can be seen in Section 5 in S2 Appendix. BSI, bloodstream infection.

in incidence were robust to using the minimum surveillance coverage values (Section 9 in S2 Appendix).

Modelling sensitivity analyses showed little effect from including regularising priors or of including samples from individuals aged 0 (Sections 10 and 11 in S2 Appendix). Our model

selection sensitivity analysis showed that even with a different structural approach, the model used in the main analysis was the preferred model, with the age$^2$ term being beneficial to model fit (Section 12 in S2 Appendix).

## Discussion

We find distinct patterns by age and sex in resistance prevalence in BSI across Europe. Surprisingly, resistance prevalence does not always increase with age nor is it higher in women, suggesting that antimicrobial exposure is not the sole driver of resistance. Instead, high spatial variation suggests that cultural factors play an important role, and differences between bacteria in terms of their natural history and pathways to infection are important to consider in tacking AMR. While there are limited previous studies looking at age and sex in the context of AMR, these do not provide their estimates of the relationships as output [21] or are limited to explorations in specific settings or bacterial species [18,27,53–55]. The only previous report considering age differences in detail focuses only on 6 bacteria-antibiotic combinations in 5-year age bands, and shows similar trends as we do [17] but did not quantify such differences nor explore them at the national, or subnational, level. The main strengths of our analyses lie in the detailed BSI data used, as we were able to use 1-year age bands, where trends may have been obscured in previous studies by high-level age groupings (e.g., [9,18,21,56]). However, we are limited by the nature of routinely available surveillance data. Also, our aim was to present the trends and hence we use a limited set of model structures. These new findings about differences by age and sex should now be considered in AMR research as they have the potential to yield new insights into AMR epidemiology and may inform the design of control measures.

Focusing on 2 critical pathogens for BSI (*E. coli* and *S. aureus*, isolated from >40% of BSI in a global survey) [51] and associated important resistances in Europe [45] and globally [52], we show substantial subnational and national variation, while demonstrating that there are age- and sex-related trends for specific bacteria-antibiotic combinations. Transmission of MRSA often occurs in healthcare settings [57] and increased contact with such settings with age could explain our observed often positive trend in resistance proportions by age. While for aminopenicillin resistance in *E. coli*, the contrasting dominant negative trend in resistance with age could be explained if, with age, more infections were endogenous and community-onset (see below). Exploring the contribution of community- versus hospital-associated infections (which we could not do with this data), as well as combinations of resistance within single isolates could test this hypothesis and explain country-level variation.

While resistance prevalence in many gram-negative bacteria often peaked at younger ages (as has been predicted for multidrug-resistant *M. tuberculosis* carriage [58]), we found little commonality in patterns between bacteria or within drug families. This suggests that the link between demographics and AMR is likely to be less driven by biological factors, but instead is driven more by cultural or behavioural factors, as alternatively we might expect similar patterns of resistance within drug families across different bacteria. Contrasting this with the biological factors that can drive increased infection risk by age suggests that there is vital information in comparing and contrasting AMR prevalence in infection spatially by age and sex to improve intervention design and antimicrobial usage. For example, understanding trends in resistance by age could lead to improved understanding of the importance of antimicrobial use variation between ages and countries [59], healthcare contact and infection prevention control practices, and even microbiological sampling that could inform both data analysis for burden and evolution understanding as well as transmission intervention potential. Comparing and contrasting antibiotic combinations used for important syndromes, such as community-acquired pneumonia, with the resistance patterns seen in common causative

organisms could point to specific age-related selective pressures and hence targets for antibiotic stewardship.

Summarising the nonlinearity of the resistance prevalence patterns by age was challenging. The level of heterogeneity was reflected in the variation in the multilevel model coefficients estimated (Table 2) and in the country-level model analysis (Fig 3). However, we found that using a simplifying index, comparing model-predicted resistance prevalence in infections in younger versus older individuals (age 1 versus 100, but robust to other age comparisons), revealed broad trends across countries. For example, showing sex-related differences that were often consistent, if varying in magnitude, across countries but were not consistent across bacteria-antibiotic combinations. With non-anonymised country data, this could be used by international agencies to support specific countries to explore drivers and prioritise the age- and sex-related targeting of interventions.

Our results could inform policy and practice in healthcare settings in a variety of ways. Firstly, understanding differences in age- and sex-related risks of infection with resistant bacteria could lead to more targeted empiric prescribing, tailored to the individual and setting, as has previously been suggested [18,20,60]. This may be particularly important in older adults that often experience more severe consequences of bacterial infection [61]. There could also be potential for improved treatment in younger adults through recommendations for the use of antibiotics which are currently limited across adults due to concerns about serious risks from resistant infections (driven by higher risk profiles for older populations) or other age-related concerns such as quinolone use linked *C. difficile* infections. Aside from more targeted age-based, and potentially sex-based, antibiotic stewardship, this analysis points to interventions that target transmission and de-colonisation strategies. For example, knowledge of higher resistance prevalence in BSI in a certain age/sex group could lead to their being a prioritisation for de-colonisation strategies [62], as well as potential candidates for single room or personal protective equipment infection prevention control interventions to prevent onward transmission.

Secondly, understanding the importance of demographic factors on AMR will support the collection and smart use of further data in this area. This is especially relevant due to the high levels of variation across settings that we identify, underpinning the need for local-level infection data collection and policies. Only this level of data would enable determination of the local sources of AMR and hence optimal targeting of interventions. While demographic data is often encouraged to be reported by countries, this is often not included in analyses, and its use is confounded by differences in the data sources and sampling practices [63]. These differences, and the subnational variation we found, highlight the need for reducing the reliance on estimates of AMR based on either single settings within a country or national averages, as done with large global estimation studies [56], as averaging across data collected from different study sites can reduce the accuracy and poorly reflect heterogeneity.

This is not only true for understanding AMR, but also BSI risks—sex differences in incidence by age could give clues to targeting this large contributor to mortality that are not commonly explored or considered in BSI epidemiology [64]. The clear higher BSI rate in men, apart from for *E. coli* infections in those aged 15 to 40, contrasts with the lack of clear sex effect in many of the resistance trends. The higher BSI levels in women aged 15 to 40 has been seen previously [65,66] and could reflect the higher urinary tract infection incidence in women [67] which are a common BSI source [68,69] and potentially reflect hormonal changes that can affect the microbiome between menarche and menopause [70]. Future work is needed to quantify the contribution of biological sex versus sex-related exposure and societal influences on both the direct and indirect risks of infection and hence AMR.

In addition to the direct implications of our findings on public health policy, understanding of the links between demographics and AMR will be foundational to a deeper understanding of acquisition routes of AMR. In our analysis, we explored demographic trends across populations but are limited in our ability to understand the mechanisms behind these trends, where further research is required. One potential avenue for such research is to explore the primary source of the bacterium causing the BSI: endogenous following long-term carriage, with a potential minor infection prior to the BSI or recent transmission. Age- and sex-related patterns in BSI source will be influenced by many factors, such as levels of contact with healthcare systems (e.g., hospital stays, previous antibiotic prescriptions [60]), individual behaviour (e.g., causes for hospital admission, rate of contact with other individuals), and inherent biology (e.g., immunosenescence, likelihood of source being a urinary tract or wound infection), as well as varying by bacteria-antibiotic combination. While the contribution of some of these factors have direct links to incidence by age (e.g., immunosenescence contributes to higher sepsis incidence with age [71]), linking these factors to proportion resistance by age is more complicated.

One theory linking the proportion of resistant infections with age is that older individuals are more likely to have weaker immune systems, and therefore are more likely to develop infections due to bacteria they are already colonised with and then enter the healthcare setting, as compared to younger individuals that would be relatively more likely to acquire a resistant bacterium through a transmission event within a healthcare setting. This would have implications for the proportion resistant by demographic characteristics for given bacteria and could be linked to changes in the microbiome with age [72]. One of the most surprising species we detected patterns for was *Acinetobacter* species, with strong age-, sex-, and subregion-related trends. This could be explained by young men being more likely to attend hospital for trauma compared to women [73], so if for this demographic population the key route to BSIs is from wound infections due to bacteria (such as *Acinetobacter* spp. [74]) acquired in hospital, this could explain the differences we observed in resistant proportion. Differences in incidence and resistance proportion could also be explained by the demographics of those who travel to areas with higher prevalence of *Acinetobacter* species [75]. Key to understanding the influence of the various factors on AMR is detailed knowledge at an individual level, as well as information on community versus hospital acquisition and antibiotic exposure, which we were unable to determine in this study. However, there are indications that for many bacteria, hospital-acquired infections are likely to have higher resistance [27] so changing contact with healthcare would be an important avenue to explore. Exploiting this demographic link could also be used to tackle and determine the key drivers of known subregional variation in resistance prevalence [45]. Our work therefore highlights the need for future research on the mechanisms of age- and sex-related AMR trends. In order to achieve this individual patient-level data, linked across primary and secondary care will be essential.

The large variation we see in age-sex-country trends in resistance prevalence is likely driven by a complex array of factors from the important high variation in antibiotic usage [35], to healthcare delivery variation [76] which can contribute to variation in healthcare practices, some of which will have sex-related impacts such as obstetric interventions [77]. Disentangling these structural differences from individual-level factors such as recent international travel [75], immune status, and comorbidities will require more detailed linked patient-level data as well as mathematical model methodology to simulate and test hypotheses as to the interaction and directionality of effect. Currently, the evidence as to the relative importance of these factors to driving the age-sex relative national differences is scarce.

Our research has several limitations. Firstly, we were unable to account for comorbidities and other syndromes of individuals, which may impact the age groups that are susceptible to

different infections. For example, cystic fibrosis patients are known to be particularly suscepti-
ble to infections of *P. aeruginosa* [78], while also being correlated with the demographics of
patients [79]. Not including such aspects may particularly bias our work on the future burden
of AMR, as the demographics of the syndromes will likely also change over time [79]. This also
highlights the need to take syndromes into account when prescribing antibiotics, as well as
demographic factors, and to record such information alongside AMR data.

Our analysis is only of European data, and not split by community or hospital-onset, and as
such may not represent universal trends that could vary in other settings, in particular where
demographic and healthcare distributions are substantially different. Our interpretation and
results are also limited by the anonymous nature of the country-linked information. We could
not report non-anonymised country-level differences at individual age and by sex as this level
of detail is not publicly available—only resistant proportion by gender and separately for age
groups 0 to 4, 5 to 18, 19 to 64, and 65+ [9]. However, reporting only the European-level asso-
ciation would mask many patterns and hence we chose to show the results by anonymised
country-level to emphasise the importance of continuing this analysis with country-specific
data in the future.

In addition, the individuals included in this data set may be biassed because of variation in
whose samples are sent to be tested for resistance—we are relying on data from routine surveil-
lance. This variation will depend on demographics and can be influenced by the age of the
individual, the severity of infection and previous failed antibiotic use, and testing guidelines,
among other factors. Understanding the decisions in sampling made by clinicians and other
healthcare professionals is a vital area of future study and may account for some of the local-
level variation we have identified [63]. Upstream of the sampling decision, it may also be influ-
enced by healthcare seeking behaviour, for example, women are more likely to seek healthcare
than men [80], and this also varies by age and potentially by country. Variation in where
within hospital settings samples are taken (ICU, A&E, etc.) may also explain some of the
national and subnational variation we observe, but need further country-specific information
to explore. Differences in sampling and processing may both create and obscure differences in
resistance prevalence between populations. By using data from TESSy, which contains only
blood and cerebrospinal fluid isolates, likely representing the most serious types of bacterial
infection [81] where the vast majority of infections will be hospitalised, these biases should be
minimal. However, this does not mean they are all sampled, and of those many samples will
test negatively for infection [82]. The EARS-NET dataset also does not include all bacteria and
all antibiotics and hence we have a limited, though clinically important, set of combinations
considered.

In terms of the analysis and modelling in this paper, we chose to limit ourselves to a "one-
size-fits-all" approach, applying the same models to each bacteria-antibiotic combination.
There is potential for models that are a better fit to the data for specific bacteria-antibiotic
combinations; however, this approach allowed us to compare model outputs across bacteria-
antibiotic combinations, as well as reducing the complexity required. Lastly, we did not
attempt to link AMR prevalence with mortality rates. This is because we did not have appro-
priate information to do this, with age-specific mortality rates and the impact of resistance on
infection being hard to estimate, with variations in baselines used (e.g., associated versus
attributable [56]). Recent estimates have found that data scarcity makes estimating relative
risks of mortality by subgroups or geographical setting difficult [21].

Future work estimating the burden of AMR and impact of interventions will need to
account for these trends by age and sex to accurately capture burden. The complexity in trends
in resistance prevalence by age and sex interact with the exponential increase in BSI incidence
with age to mean that often, the elderly population, especially men, would still be expected to

suffer more infections with resistant bacteria. How this collides with the global shift to older populations [2] and the impact this will have on public health burden as well as AMR spread should be a research priority. Future work is also needed to explore the national and subnational variation using age- and sex-disaggregated antibiotic usage data which is not currently widely available, as well as other national policy differences in infection control, antibiotic stewardship, and demographic trends. Fundamental differences in healthcare contact by sex related to pregnancy and chronic disease patterns also need to be explored in relation to driving both selection and transmission differences in resistance prevalence.

In 2018, the WHO asked "Is the impact of AMR the same for everyone? Do any groups in society face greater or different risks of exposure to AMR or more challenges in accessing, using and benefiting from the information, services and solutions to tackle AMR? If yes, who, why, and what can we do about it?" [30]. In this paper, we go some way to addressing these questions by quantifying how AMR prevalence in BSIs across Europe varies by age and sex, as well as identifying variation at the local level. We do not wish to oversimplify any trends in AMR by age or sex—risk factors, previous prescribing as well as contact with high-risk transmission settings such as hospital or long-term care facilities will all influence individual-level risk of AMR infection. However, our ecological analysis shows the substantial interactions of age and sex with AMR, and we therefore encourage their inclusion in future data collection and research studies to improve health outcomes across the spectrum of AMR.

## Supporting information

**S1 Appendix. Additional methods.**
(DOCX)

**S2 Appendix. Additional results and sensitivity analysis.**
(DOCX)

**S3 Appendix. Bacteria-antibiotic-specific model results.**
(PDF)

## Acknowledgments

We are grateful for all the work done by the staff of the participating clinical microbiology laboratories and of the national healthcare services that provided data to EARS-Net. We also thank Dr. Sam Abbot, Dr. David Hodgson, and Dr. Tim Russel for discussing model fitting complexities with us.

## Disclaimer

The views and opinions of the authors expressed herein do not necessarily state or reflect those of the European Centre for Disease Prevention and Control (ECDC). The accuracy of the authors' statistical analysis and the findings they report are not the responsibility of ECDC. ECDC is not responsible for conclusions or opinions drawn from the data provided. ECDC is not responsible for the correctness of the data and for data management, data merging, and data collation after provision of the data. ECDC shall not be held liable for improper or incorrect use of the data.

## Author Contributions

**Conceptualization:** Gwenan Mary Knight.

**Data curation:** Naomi R. Waterlow, Gwenan Mary Knight.

**Formal analysis:** Naomi R. Waterlow, Gwenan Mary Knight.

**Funding acquisition:** Gwenan Mary Knight.

**Investigation:** Naomi R. Waterlow.

**Methodology:** Naomi R. Waterlow, Ben S. Cooper, Julie V. Robotham, Gwenan Mary Knight.

**Software:** Naomi R. Waterlow, Gwenan Mary Knight.

**Supervision:** Gwenan Mary Knight.

**Visualization:** Naomi R. Waterlow, Gwenan Mary Knight.

**Writing – original draft:** Naomi R. Waterlow.

**Writing – review & editing:** Ben S. Cooper, Julie V. Robotham, Gwenan Mary Knight.

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
