## [Editor Report · Decision Letter 0]

21 Sep 2023

Dear Dr Knight, 

Thank you for submitting your manuscript entitled "How demographic factors matter for antimicrobial resistance – quantification of the patterns and impact of variation in prevalence of resistance by age and sex." for consideration by PLOS Medicine.

Your manuscript has now been evaluated by the PLOS Medicine editorial staff and I am writing to let you know that we would like to send your submission out for external peer review.

Please re-submit your manuscript within two working days, i.e. by Sep 25 2023 11:59PM.

Kind regards,

Pippa

Philippa Dodd, MBBS MRCP PhD

PLOS Medicine

---

## [Decision Letter · Decision Letter 1]

23 Oct 2023

Dear Dr. Knight,

Thank you very much for submitting your manuscript "How demographic factors matter for antimicrobial resistance – quantification of the patterns and impact of variation in prevalence of resistance by age and sex." (PMEDICINE-D-23-02731R1) for consideration at PLOS Medicine. 

[LINK]

In light of these reviews, I am writing to let you know that we would be happy to consider a revised version that addresses the reviewers' and editors' comments. Obviously we cannot make any decision about publication until we have seen the revised manuscript and your response, and we plan to seek re-review by one or more of the reviewers. 

We expect to receive your revised manuscript by Nov 13 2023 11:59PM. Please email us (plosmedicine@plos.org) if you have any questions or concerns.

We look forward to receiving your revised manuscript. 

Sincerely,

Philippa Dodd, MBBS MRCP PhD

PLOS Medicine

plosmedicine.org

COMMENTS FROM THE EDITORS

GENERAL

Please respond to all editor and reviewer comments detailed below in full.

Please include line numbers starting at 1 on the title page and in continuous sequence throughout thereafter.

The editorial team agree that your study could be of significant value to the field. However, in its current form it does not flow well and is in places somewhat inaccessible to the reader which detracts from its impact. Please pay careful attention to the presentation of your manuscript, in particular the presentation of your methods and results when you revise. Specific comments are detailed below pertaining to each sub-section. We encourage you to review published articles on our website here https://journals.plos.org/plosmedicine/ and to refer to our guidelines on revising your manuscript which can be found here https://journals.plos.org/plosmedicine/s/revising-your-manuscript 

COMPETING INTERESTS

All authors must declare their relevant competing interests per the PLOS policy, which can be seen here:

https://journals.plos.org/plosmedicine/s/competing-interests

For authors with ties to industry, please indicate whether any of the interests has a financial stake in the results of the current study.

TITLE

Please revise your title according to PLOS Medicine's style. Your title must be nondeclarative and not a question. It should begin with main concept if possible. "Effect of" should be used only if causality can be inferred, i.e., for an RCT. Please place the study design ("A randomized controlled trial," "A retrospective study," "A modelling study," etc.) in the subtitle (ie, after a colon).

ABSTRACT

Please structure your abstract using the PLOS Medicine headings (Background, Methods and Findings, Conclusions).

Please combine the Methods and Findings sections into one section, “Methods and findings”.

Abstract Background: Please ensure that the background constitutes only one paragraph which includes context of the study importance. The final sentence should clearly state the study question. Please remove details of the statistical approach/study design ad place in the methods and findings section instead.

Abstract Methods and Findings:

Please clearly describe how you addressed your question/hypothesis.

Please include details of the databases that you used to leverage data that you present/describe.

Please include the study design, population(s) and setting(s), number of participants and/or samples included in your study, years during which the study took place, length of follow up, and main outcome measures. 

Aggregate demographic details of the population studied would also be helpful (age range and mean and stratified according to sex) please include here instead of later in paragraph 2.

Please ensure that you detail the full names of bacteria, Escherichia coli instead of E. Coli for example.

Results paragraph 2: suggest instead, ‘aged between 1 and 100 years’ but as above, suggest including this information earlier along with the description of the study population. Please also see reviewer comments below regarding age cut-off.

Results paragraph 2: ‘We explore our results in greater depths for two of the most clinically important bacteria–antibiotic combinations.’ would be better placed with the description of your methodological approach instead of the results section. Some additional clarification as to why these are ‘the most clinically important’ would be helpful to the non-expert reader.

Results paragraph 2: ‘At the country-level, the patterns are highly context specific with national and subnational differences accounting for a large amount of resistance variation…’ this is very vague – what, broadly, are the different contexts you refer to? Similarly, what are some examples of the differences you refer to? Please elaborate for clarity.

The following sentence, ‘This diverges from the known, clear exponential increase in infection incidence rates by age…’ is this reference to data published elsewhere? If so, please remove. In the abstract, please refrain from including reference to data other than that generated by your study.

Please ensure that all numbers presented in the abstract are present and identical to numbers presented in the main manuscript text.

Please clearly define all statistical information for the reader. What does the numerical value, ‘…~0.46…’ represent? And, similarly what does the preceding symbol represent? Please clarify and define for the reader. 

Please define CI at first use for the reader.

Please quantify the main results with 95% CIs and p values. 

When reporting p values please report as p<0.001 and where higher as p=0.002, for example. If not reporting p values, to help facilitate transparent data reporting, please clearly state the reasons why not.

When reporting CIs please use commas as opposed to hyphens to separate upper and lower bounds as the latter can introduce confusion especially when reporting negative values.

Please include any important dependent variables that are adjusted for in the analyses.

In the last sentence of the Abstract Methods and Findings section, please describe the main limitation(s) of the study's methodology.

Abstract Conclusions:

As above, the conclusions could be slightly more nuanced regarding ‘context specific patterns’, please elaborate.

Please ensure that you address the study implications without overreaching what can be concluded from the data; the phrase "In this study, we observed ..." may be useful.

Please interpret the study based on the results presented in the abstract, emphasizing what is new without overstating your conclusions.

Please avoid vague statements such as "these results have major implications for policy/clinical care". Mention only specific implications substantiated by the results.

Please avoid assertions of primacy ("We report for the first time....")

AUTHOR SUMMARY

At this stage, we ask that you include a short, non-technical Author Summary of your research to make findings accessible to a wide audience that includes both scientists and non-scientists. The authors summary should consist of 2-3 succinct bullet points under each of the following headings:

• Why Was This Study Done? Authors should reflect on what was known about the topic before the research was published and why the research was needed.

• What Did the Researchers Do and Find? Authors should briefly describe the study design that was used and the study’s major findings. Do include the headline numbers from the study, such as the sample size and key findings. 

• What Do These Findings Mean? Authors should reflect on the new knowledge generated by the research and the implications for practice, research, policy, or public health. Authors should also consider how the interpretation of the study’s findings may be affected by the study limitations. In the final bullet point of ‘What Do These Findings Mean?’, please describe the main limitations of the study in non-technical language.

Author Summary should immediately follow the Abstract in your revised manuscript. This text is subject to editorial change and should be distinct from the scientific abstract. Please see our author guidelines for more information: https://journals.plos.org/plosmedicine/s/revising-your-manuscript#loc-author-summary

INTRODUCTION

Please indicate whether your study is novel and how you determined that. If there has been a systematic review of the evidence related to your study (or you have conducted one), please refer to and reference that review and indicate whether it supports the need for your study.

Para 1: sentence beginning, ‘The higher burden…’ please provide a reference for this statement.

Para 2: ‘The recent WHO reports also have no presented analysis of how resistance prevalence varies by these demographic factors (8,8).’ Suggest ‘not’ instead of ‘no’. Please also check referencing/citations here, ‘…(8,8).’ 

Penultimate paragraph: ‘…is key to understanding the complexity of AMR interventions.’ Not sure this accurately defines the point you are making. Perhaps instead, ‘…is key to understanding the nuanced interventions required to tackle AMR.’ or similar.

Final paragraph: opening sentence suggest, ‘Here, we use a large dataset of routinely collected information on bloodstream infections to explore trends in prevalence of antibiotic resistance and infection by age and sex across Europe.’ 

Please remove the final sentences beginning with ‘It is vital…’. You could consider including these in the discussion section of your manuscript but, they are very subjective thus will require revision. We encourage you to include only objective statements substantiated by the data generated by your study.

METHODS and RESULTS

A lot of nuanced information regarding what you did and why is missing from these sections. Please report the number of patients, samples, etc and dates of recruitment/inclusion, and account for all methods used in your study. Please revise in line with the specific comments detailed below.

You state that ‘Countries were anonymized using a random 3-letter code, which is used throughout the paper’ why was this done? It seems unintuitive to present your results at country level and then to anonymize the countries. Please also see comments from the Academic Editor and the reviewer in relation to this point.

Methods opening paragraph: this would be better placed as the opening paragraph to your results section.

Sentence beginning: ‘All code is available…’ and ‘Patient level data is available…’ please remove these statements from the main manuscript and include only in the manuscript submission form. In the event of publication this information will be compiled as metadata.

Methods paragraph 2: please delete. Please include only precise details of your methodological approach under appropriate sub-headings.

Data paragraph 1: please define ‘ECDC’ at first use for the reader. In the opening paragraph please clearly state the years which data were extracted for analysis i.e. from when to when? 

In the appendix you present a vast amount of information regarding final participant inclusion as well as in respect to the isolates and stratified by country (albeit anonymized). But, I couldn’t see any detailed information either in the main manuscript or in the appendix regarding the population that the isolates of interest came from. The reader has to do a lot of jumping about to find nuanced information which should be readily available and accessible. Please revise accordingly.

Results opening paragraph: please include a paragraph which summarizes your study population including the total number, basic aggregate demographic details as well as a table of baseline characteristics of the total population and stratified by country in the main manuscript. You could also include relevant information pertaining to the isolates of the final population included.

Resistance prevalence: please remove, ‘…supported clear benefits of including age in predictions of resistance in infection…’ which would be better place in the discussion as it is interpretive. 

‘Model analysis: examples’ and ‘Model analysis: general results’ – these are rather vague and uninformative. Please revise. What do you mean by ‘examples’ and ‘general results’? ‘Model analysis’ is poorly representative of what you present. What are you showing the reader? Please revise to include more informative sub-headings that clearly inform the reader of the content that follows.

Sensitivity analyses: ‘Analysis…revealed large differences between countries across Europe’ please include the full data without anonymizing the countries. Please ensure that you provide detailed discussion of the potential reasons for the differences in observed outcomes in the discussion section. 

When reporting outcomes please quantify results with 95% CIs and p values. When reporting p values please report as p<0.001 and where higher, the exact p value as p=0.002, for example. If not reporting p values to help facilitate transparent data reporting please clearly state the reasons why not.

FIGURES

Throughout, please consider avoiding the use of green and/or red to make your figures more accessible to those with colour blindness.

Throughout you use the abbreviation ‘CI’ to refer to both confidence intervals and credible intervals. Please justify. Please ensure that CI is defined in all captions/footnotes for the reader.

Throughout please indicate whether analyses are adjusted and if so, which factors are adjusted for. Where adjusted analyses are presented please also present unadjusted analyses for comparison.

Figure 1 – It isn’t at all obvious that the size of the points on the plots differ, largely due to the size of the individual graphs and, in addition no dots are black which makes the legend somewhat redundant and rather confusing please revise. The legend to depict male and female is also very confusing. The colours are obscured by the colours of the plots. Suggest using different lines to represent male and female and removing the colours as it is impossible to differentiate between the 2 groups in some instances. When looking at the ‘columns of graphs’ what do the double headers refer to? Presumably the top graph is antibiotic class and the bottom the specific antibiotic example? Please clarify. And if, indeed if I am correct suggest either labelling the graphs separately or clearly defining this for the reader in the caption. The use of ‘rows’ and ‘columns’ to describe different graphs is somewhat unconventional, please revise.

Figure 2 – as above suggest using different black solid/dashed lines here also to represent male and female populations. It is difficult to appreciate where upper and lower bounds of CIs end when using shading. Is there a better way to present these data? Please revise. For ‘Pseudomonas aeruginosa’ the colour shading is too pale to be clearly visible, suggest revising your choice of colour palate.

Figure 3 and 4 – please define ABO, NSL, PUB, KVX, BPQ in the footnote (if you type ‘country ABO’ into the Google search engine the headline is blood type by country…). In the footnote of figure 4 you state that these letters ‘are random and anonymized’ – why? This is not clear in your methods section. Please clarify and revise. What does ‘CI’ refer to in the footnote? Please define for the reader. In part A, ‘upper 95%’ an ‘lower 95%’ what? Please define. You could combine these into one column, separating with a comma, and completely define the data in the column header (confidence interval or credible interval) to improve accessibility. Is the male and female legend required considering the labels on the graphs? The grey shaded dots for sample size are misleading as there are no grey dots in these graphs. Please revise. Again, please refrain from referring to graphs as ‘columns’ you are not describing a table.

Figure 5 – please clearly define the meaning of the dots and lines for the reader in the footnote/caption.

TABLES

Please provide a table showing the baseline characteristics of the study population.

DISCUSSION

Please include a wider and more nuanced discussion of the differences that you observe, as per the reviewer and academic editor comments.

ACKNOWLEDGEMENTS

Please move the declaration to the competing interest section of the manuscript submission form. Please define ECDC. Is an author affiliated to the ECDC? If so, please clearly state this by detailing their initials. 

REFERENCES

Please ensure that referencing follows our guidance which can be found here https://journals.plos.org/plosmedicine/s/submission-guidelines#loc-references

Please ensure that all journal name abbreviations are those found in the National Center for Biotechnology Information (NCBI) databases. 

For in-text reference callouts please place citations in square brackets and preceding punctuation as follows, ‘[1,3,].’

Please ensure all web references include an ‘Accessed [date]’ as opposed to ‘Cited [date]’.

SUPPORTING INFORMATION

Please cite your Supporting Information as outlined here: https://journals.plos.org/plosmedicine/s/supporting-information

In the published article, supporting information files are accessed only through a hyperlink attached to the captions. For this reason, you must list captions at the end of your manuscript file. You may include a caption within the supporting information file itself, as long as that caption is also provided in the manuscript file. Do not submit a separate caption file.

Please ensure that all abbreviations used in figures are clearly defined for the reader including those used to depict statistical information.

APPENDIX 1

Supplementary figure 5 – it is impossible to differentiate the data pertaining to individual bacteria when grouped in this way. The bars are far too small. Please revise.

Supplementary figure 6 – as above the graphs are too small to be easily accessible to the reader. Please revise. As for the main manuscript please refrain from describing the graphs as columns and rows.

Please follow the referencing format as outlined above for the main manuscript. 

APPENDIX 2

Figures - The same problems identified above (appendix 1) and in the main manuscript also apply here. Please revise in line with the previous comments to improve clarity and reader accessibility.

Table 1 – as for the main manuscript, please separate upper and lower bounds of CIs with commas as opposed to hyphens which introduce confusion when reporting negative values.

COMMENTS FROM THE ACADEMIC EDITOR

This has potential to be a strong manuscript, and of high public health and clinical importance. However, I agree with you and the reviewers that major revisions are required. I would be happy to see a revised version resubmitted.

Specific comments that the authors should address:

1) Greater reflections and discussion on the public health importance of variation in incidence and prevalence by age, sex, and country. What do these findings imply needs to be done to reduce morbidity, mortality, and improve health? Right now, discussion of these issues is somewhat superficial, focusing on "context specific interventions" without mentioning what these could be, and what their likely impact would be. What do the authors recommend policymakers do with these results?

2) Needs a strong justification for why countries are anonymised. Right now, there is little value in the country-stratified estimates, as readers will not be able to interrogate the possible reasons for differences in trends between countries. So, recommend presenting country names, or if not possible (and I am not clear why it would not be possible), remove these analyses.

3) I don't understand why the analysis comparing people aged 0 and 100 years is included, or deemed to be interesting, particularly given the U-shaped distribution found for many antibiotics, and the very small number of 100 year olds. Recommend either providing a clear justification for why this is felt to be important (and is robust to extrapolation beyond the available data), or rework/remove this analysis.

4) I agree with several reviewers that figures are overly complex, and difficult to interpret due to data overload and design choices. Please simplify and reduce the number of figures to convert selected important findings.

5) Greater discussion of the possible underlying causes of age-sex-country trends in prevalence and incidence is required. What do the authors think are the underlying causes of variation here? What evidence supports this? What specific research is now needed to better understand the drivers of these patterns?

6) Thanks for providing code. The "base" model is described as having "flat" priors. Looking at the code in the GitHub repo, I don't think that is correct. Models specify that the default `brms` priors are used, which are not flat. Given the large amount of data available, I am not clear why "flat" priors would be a useful modelling choice here, and this obviates the decision to use a Bayesian approach to analysis. Recommend the authors carefully review a) prior selection and b) terminology.

Comments from the reviewers:

Reviewer #1: See attachment

Michael Dewey

Reviewer #2: This manuscript contains an analysis of the AMR prevalence and incidence patterns in Europe between 2015 and 2020, stratified by sex and age. It is clearly written and the exposition is of a high calibre. The importance of the work is also clear, as sex and age have not been systematically considered as risk factors for AMR in previous studies, except possibly the one that the authors cited. There are certain methodological concerns that should be addressed before it can be accepted, that except for the first two are mostly minor.

The main concern has to do with the multiple hypothesis testing issue. Although this issue is less severe in the case of a multilevel Bayesian analysis, as was used in this manuscript, there is a need to explain why that is the case. A good, albeit somewhat dated, reference might be "The Statistical Crisis in Science" by Gelman and Loken, which highlights both pitfalls and ways around them. In particular, a specific concern is the inclusion of age^2 as an explanatory variable - the rationale for this seems unclear unless the authors believed from the start that the age dependency follows a quadratic curve (but their discussion of prevalence of infection increasing with age suggests otherwise). At a minimum, an additional sensitivity analysis without this variable would be recommended.

Secondarily, I am not entirely convinced that the uninformative priors used here are appropriate; the alternative of using a N(0,1) prior on the fixed-effect parameters could be described as a weakly regularising prior, but perhaps a strongly regularising prior could be used instead, given that nearly 10M datapoints are included in the study. There is a helpful discussion of the tradeoffs of regularisation via different priors at this link among several other places: betanalpha.github.io/assets/case_studies/bayes_sparse_regression.html

Other minor comments follow below.

1) In the main equation, if i is indeed the grouping level, then (according to the previous paragraph) it already includes country and lab, so are the indices c and l superfluous? Perhaps they could be omitted in the top equation and replaced by c(i) and l(i) in the bottom one when specific aspects of the grouping need to be referred to? Alternatively, if i refers to something else, please clarify its exact meaning in the equation.

2) The residual error epsilon should be indexed by i, not t (unless the error is only time-dependent)

3) Please clarify that the interaction variable age * sex uses a 0-1 coding for sex (otherwise the product is meaningless), and specify which one is 0

4) The male-female colour distinction is hardly visible in Figure 1; either remove it altogether or enhance the contrast by zooming into the relevant parts of the plot

5) The fq_pseudo_R should be reserved for P. aeruginosa (Table 1), unless the exact same definition of fluoroquinolones was used for other species, in which case please state this explicitly; additionally, it may be worthwhile combining tables 1 and 2

6) In the introduction, the same citation is used twice, appearing as (8,8)

7) samples from individuals aged -> samples from individuals aged 0

8) of those there are many samples -> of those many samples

9) The minimum of 1.4% untested is inconsistent with the maximum of 98.7% tested (likely due to rounding issues) in Table 1

10) and are also likely to have a BSI -> isolates do not have a BSI, patients do (also spell out the BSI acronym the first time it is used if you have not already done so)

11) Southern and Eastern generally -> Southern and Eastern Europe generally

12) "The patterns are still more similar" -> the first half of this sentence contradicts the second half

13) Philosophical sensitivity analysis -> this sounds slightly unusual, although the intention is clear; structural (or methodological?) sensitivity analysis would probably be a better name

Reviewer #3: This is an important analysis and provides useful information. The work has been robustly conducted and is reported clearly and concisely. If published in PLOSmedicine it will be highly cited

I have a series of minor suggestions for the authors to consider.

Title : Quantification of the patterns and impact of variation in prevalence of resistance by age and sex

I am not sure the paper really addresses impact. The authors may want to consider removing this from the title

Introduction

Line 5 "rarely quantified" - no ref

Para 2 line 7 (8,8) needs correcting

Para 4 "Despite awareness of the importance of sex for many risk factors for infectious diseases (e.g. HIV) and

bacterial infections such as urinary tract infections,…" I think this sentence is too general (sex links more widely to poor outcome from infection - and almost every other acute illness but specifically to risk of specific infections such as UTI) to be helpful and HIV is such a different disease. Suggest something like Despite sex being a well-established risk factor for specific bacterial infections such as UTI …

Para 4 line 4 "studies of single bacteria" feels odd. Bacteria aren't really single. Suggest " infections caused by specific bacterial pathogens"

Para 5 "by

"factors such as governance and deprivation level" do you mean linked to or associated with? What is meant by governance leve, this isn't clear to me at least.

Methods

These rely on EARs-NET data which are I think opportunistic, by which I mean labs submit data based on the clinical samples they receive. Differences in sampling and processing will therefore have the potential to create apparent differences between countries obscuring presence or absence of real differences. 

Para 3. In our main analysis we exclude individuals aged 0, due to their stark difference in immune dynamics and

contact patterns, but run a sensitivity analysis including them.

This feels rather stark. Why 1 not 2? Or 10? Is there a better rationale or precident you can reference?

Results 

"At the European level, there were clear non-linear differences in the prevalence of resistance in

infection by age and sex for different bacteria-antibiotic combinations (Figure 1)."

Figure 1 conveys a lot of information and could be made clearer

V hard to distinguish sex with colour - suggest you make one line dotted and the other continuous. 

Took me ages to note that the rows are gram negative and positive species respectively 

Suggest you use these as headings rather than on the right y Axis so with a third top band to each of the 8 rows

Why have you separated out amikacin from (other presumably?) amino glycosides

Ansamycin isn't a widely used term I think - rifamycins would be better?

I am not sure I agree with some of the text description of the trends shown in figure 1. There are more messages to draw out I think…

The age-associated patterns varied more within drug-families than within certain bacteria (patterns within each colour are more different than within each row of Figure 1)

What about the penicillin and macrolide data for strep pneumo - you don't have data for these deugs for other organisms and they are v like eachother but very different from the other Strep pneumo graphs. These are the two first line drugs for CAP in most systems probably

"Sex has little impact on many of the age-related trends except for Acinetobacter species at younger ages, and E. coli and Klebsiella at higher ages"

Resistance among Acinetobacter peaks in young adult males across all classes and similar peaks are seen for klebsiella but not E coli. Pseudomonas resistance peaks in young adults of both sexes. There are no apparent gender differences for the gram positive species you have looked at. These data make me wonder about differences in healthcare contact (amount and nature) 

I suspect that in Men and women have different quantitative and qualitative hospital contact at different points in their lives. In many healthcare systems women will have healthcare contact in relation to pregnancies. Young men may have different forms of contact - trauma for example. Chronic disease patterns will vary too. Are there high-level data e.g. on overnight hospital stays by age and gender and icu overnight stays? 

If you hypothesize that biological sex rather than sex-related exposure is important in determining the differences you see then would you expect to see changes around the times of menarche and menopause? This is clear for E coli related presumably to UTI but otherwise?

Incidence

Figure 2 E coli - this is where you clearly see the impact of biological sex - increasing above males at menarche and converging again post menopause

Resistance prevalence

Why do age and age2 flip from being positive to negative? Understanding this paragraph requires quite a high level of statistical understanding. It would be worth a few lines of additional text to "unpack" the implications of this analysis.

Table 1 - why does high-level aminoglycoside appear twice for E. faecalis?

Model analysis

Particularly for aminopenicillin resistance in E coli the dramatic differences between countries begs some explanation which is difficult with complete anonymity. Are there other analyses you could undertake to explore hypotheses eg grouping by European region? Population level data on use of aminopenicllins per capita or national policy information. 

Discussion

"We find no universal trends, with variation in age and sex patterns across specific bacteria-antibiotic combinations and across national and sub-national contexts, suggesting that cultural factors dominate biological ones.

The exception being E. coli BSI in women? Worth highlighting this?

Subnational is variably hyphenated as sub-national.

"Transmission of MRSA often occurs in healthcare settings (52) and increased contact with such settings with age could explain our observed often positive trend in resistance proportions by age. "

This is a significant generalization and could explain why a small number of countries don't show the increase in MRSA with age. Are these areas with different epidemiology, less nosocomial and more community MRSA. You could explore this by looking at MRSA resistance to e.g. quinolones? 

"This may be particularly important in older adults, that often experience more severe consequences of bacterial infection (56)."

They are also likely at greatest risk from AMR and I would argue that it may be more important in younger adults who may be at little risk of harms from AMR but in whom we avoid drugs out of concerns about AMR that could be beneficial. My example would be the avoidance of quinolone antibiotics driven largey by concerns about C difficile infection which is a disease which only affects older adults. So we treat younger adults with UTI using with agents like betalactams that are probably inferior in other ways. 

" When considering empiric prescribing guidelines, it is vital to not just consider the impact of prescribing the most appropriate drug, but also the impact of delaying prescription until more information is available, which may have catastrophic consequences in severe infections (Girard and Ely 2007)." 

This is true but it doesn't seem very relevant - and the reference isn't cited correctly for some reason. Delaying is just one strategy so this feels like a fragment of unnecessary detail. Suggest you just delete it.

[LINK]

---

## [Decision Letter · Decision Letter 2]

21 Dec 2023

Dear Dr. Knight,

Thank you very much for re-submitting your manuscript "Variation in antimicrobial resistance prevalence in bloodstream infection by age and sex: An analysis of European data" (PMEDICINE-D-23-02731R2) for review by PLOS Medicine.

I have discussed the paper with my colleagues and the academic editor and it was also seen again by 2 reviewers. I am pleased to say that provided the remaining editorial and production issues are dealt with we are planning to accept the paper for publication in the journal.

[LINK]

We look forward to receiving the revised manuscript by Dec 28 2023 11:59PM.   

Best wishes,

Pippa

Philippa Dodd, MBBS MRCP PhD

PLOS Medicine

plosmedicine.org

COMMENTS FROM THE ACADEMIC EDITOR

I have read through the manuscript this morning, and it is much improved. Particularly the much more detailed focus on public health and future research implications of their findings.

My remaining comments are mostly minor and relate to the Figures, which I think could still be simplified and improved for clarity. In principle, I would be happy to progress to acceptance.

1) Figure 1. There is really no need to show the points here. As there is so much data, they obscure the fitted curves and uncertainty intervals for sex-specific trends. I would either a) removal the points altogether, and just show the sex-specific curves and uncertainty intervals, or b) make the points black, increase the alpha substantially, and vary the shape by sex. For black panels, need to add to the footnote to state that "no data available" (rather than the current ambiguous phrasing). Footnote still states "confidence interval" - is this correct? Should it be "credible interval"?

2) Figure 5. I struggled with interpreting this figure. E.g. in Panel A, why so some bacterial have only two estimates, and some have more (e.g. up to 6). i.e. the points and uncertainty intervals clear map to something, but not clear what it is. Should be distinguished by e.g. a different shape of the point. Change in proportion label in X-axis: should be clearly stated whether this is relative change in proportion, or percentage point change.

3) It is disappointing that the ECDC would not give permission for identification of countries, as this does weaken the insights that can be drawn from the results. I am sure readers will pick up on this too. I think as currently presented, this is fine, but from a journal perspective, I wonder if getting written evidence that ECDC have indeed prohibited release of these aggregate data might protect against future questions from readers?

COMMENTS FROM THE EDITORS

GENERAL

Thank you for your considerate and detailed responses to previous editor and reviewer comments. Please see below for further comments which we require you address prior to publication.

In respect of the Academic Editor’s 3rd comment above, please include relevant supporting information. 

TITLE

Thank you for revising your title, we suggest the following, “Prevalence and characteristics of antimicrobial resistance in bloodstream infections in 29 European countries: An observational study”

ABSTRACT

Please detail that your data is representative of 29 countries.

Line 33 – ‘6,862,577 susceptibility results’ please clarify how many participants this pertains to.

Line 35 – ‘with a similar age distribution in both sexes’ – I couldn’t see the age range detailed, please include.

Line 39 – please detail the bacteria-antibiotic combinations you refer to here.

Line 41 – please remove the limitations to the end of the methods and findings section.

Line 45 – would an example of a bacterial species you refer to also be helpful (perhaps one that is targeted by fluoroquinolones?)

Lines 52-54 – please quantify the percentages (as above, it is not mentioned in the abstract how many countries your study investigates).

AUTHOR SUMMARY

Thank you for including an author summary.

Line 79 – suggest beginning, ‘We fitted…’

Lines 81-83 – suggest combining as follows, ‘Distinct patterns in resistance prevalence by age were observed across Europe for different bacteria.

Line 84 suggest – ‘[Female] sex was most strongly associated with E. coli and K. pneumoniae resistance [in older age groups], and at younger ages for Acinetobacter species.’ Which sex most strongly associated with which resistant bacteria (male or female)? Please include details as indicated. Please also detail the full names of the bacteria. Is resistance of E coli and K pneumoniae noted mostly in older people? If so, please detail as suggested above.

Lines 87 onwards are a little repetitive and somewhat vague. And, although earlier you refer to sex and age differences here you comment only on the implications of age and as such these remarks seem incomplete. Please revise to provide a more nuanced and balanced interpretation of your findings.

In the final bullet point of ‘What Do These Findings Mean?’, please describe the main limitations of the study in non-technical language.

METHODS and RESULTS

As for the abstract please clarify and provide details of the number of participants the susceptibility results pertained to.

TABLES

Table 1 – PLOS Medicine requests that means and SDs are reported, please include. If there a specific reason that you choose to report the median then please give details. As above, please clarify and provide details of the number of participants the susceptibility results pertained to.

FIGURES

We agree with the Academic Editor regarding the presentation of your figures which we think could be improved. Please revise.

DISCUSSION

Please revise the opening paragraph of you discussion such that it begins with a short, clear summary of the article's findings; followed by what the study adds to existing research and where and why the results may differ from previous research; strengths and limitations of the study; implications and next steps for research, clinical practice, and/or public policy; one-paragraph conclusion. 

Line 672 – please place a header which reads, ‘Disclaimer’.

SOCIAL MEDIA

To help us extend the reach of your research, please detail any X (formerly Twitter) handles you wish to be included when we tweet this paper (including your own, your coauthors’, your institution, funder, or lab) in the manuscript submission form when you re-submit the manuscript.

COMMENTS FROM THE REVIEWERS:

Reviewer #1: The authors have addressed all my points

Michael Dewey

Reviewer #3: Thank you for your thorough approach to the reviewers' comments. The manuscript is considerably strengthened. I have no further comments.

[LINK]

---

## [Editor Report · Decision Letter 3]

22 Jan 2024

Dear Dr Knight, 

On behalf of my colleagues and the Academic Editor, Professor Peter MacPherson, I am pleased to inform you that we have agreed to publish your manuscript "Antimicrobial resistance prevalence in bloodstream infection in 29 European countries by age and sex: an observational study" (PMEDICINE-D-23-02731R3) in PLOS Medicine.

Thank you for your query regarding the abstract length, there is no need to further edit this in respect of the word count.

Prior to publication, please address the following:

1) Discussion line 611 onwards - please provide a link to the dashboard page (as you previously suggested) to show that the available data is not disaggregated at country level. 

2) Please also include a line or two, for the benefit of the readers, detailing your attempt to obtain these data. We thank you for your diligence regarding this point.

PRESS

Thank you again for submitting to PLOS Medicine, it has been a pleasure handling your manuscript. We look forward to publishing your paper. 

Kind regards,

Pippa 

Philippa Dodd, MBBS MRCP PhD 

PLOS Medicine

pdodd@plos.org